Subject Areas:
civil engineering/fractals/analysis

Keywords:
backfill, pore structure, fractal dimension, flowability, mechanical properties, correlation

Author for correspondence:
Qifan Ren
e-mail: qifanren@csu.edu.cn

# Trans-scale relationship analysis between the pore structure and macro parameters of backfill and slurry

Jianhua Hu, Qifan Ren, Xiaotian Ding and Quan Jiang

School of Resources and Safety Engineering, Central South University, Changsha 410083, People's Republic of China

  JH, 0000-0003-1293-6829; QR, 0000-0002-9814-834X;
XD, 0000-0002-2883-9479

The characteristics of the porous structure of backfill are directly related to the macro parameters of the flowability of the filling slurry and the mechanical features of the backfill, which are fundamental to the study of multiscale mechanics of backfill. Based on the geometry and fractal theory, scanning electron microscopy images of backfill were analysed by image analysis methods such as OTSU and box counting. The fractal dimension of the pore structure was calculated. By quantitatively characterizing the pore structure, the trans-scale relationships between the fractal dimension of the pore structure and the macro parameters of the filling slurry were established in terms of equilibrium shear stress (ESS) and equilibrium apparent viscosity (EAV). In addition, the correlations between the fractal dimension and macro parameters of backfill were obtained in terms of uniaxial compressive strength (UCS), water content (WC) and porosity. The influence of the microstructure on the macro parameters was discussed. The results show the following: (i) the fractal dimension of the backfill pore structure can characterize the complexity of the structure; (ii) the fractal dimension of the pore structure is negatively correlated with the ESS and EAV of the filling slurry. The UCS of the backfill is positively correlated with the flowability parameter; (iii) the fractal dimension of the pore structure has a certain correlation with some macro parameters of the backfill, i.e. the fractal dimension is negatively correlated with the UCS and positively correlated with the WC and (iv) the linear correlations between the pore fractal dimension and UCS and WC are established. The correlation coefficient between the fractal dimension and UCS has an $R^2$ value of $-0.638$, while the corresponding value of the fractal dimension and WC is 0.604. UCS and WC can be predicted by the fractal dimension of pores.

# 1. Introduction

To reduce the cost of cement in concrete and to manufacture cement for various special purposes, many scholars worldwide have conducted extensive work on cement substitute materials and concrete additives. Siddique [1] studied the effects of volcanic ash on the consistency, setting time, workability, compressive strength and electrical resistivity of cement paste and mortar. De Weerdt et al. [2] found that the presence of 5% limestone in concrete led to an increase in the volume of hydrates, as visible in the increase in chemical shrinkage and an increase in the compressive strength. Kupwade-Patil et al. [3] investigated the effectiveness of using volcanic ash along with silica fume as a partial replacement for Portland cement. Celik et al. [4] reported the composition and properties of highly flowable self-consolidating concrete mixtures made of high proportions of cement replacement materials such as fly ash and pulverized limestone instead of high cement content. Al-Kheetan et al. [5–7] introduced crystalline material along with a curing compound in fresh concrete to protect and extend service life and developed hydrophobic concrete by adding a dual-crystalline admixture during the mixing stage. However, there are few materials used in mine backfilling, and research on these materials is limited. Hu et al. [8] assessed stone powder as a replacement for cement in cemented paste backfill (CPB) and studied the strength characteristics and reaction mechanism. The results showed that the strength of the backfill was greatly reduced at an early stage and slightly reduced in the final stages, while the trans-scale characteristics between the pore structure and macro parameters still need to be studied.

For CPB, slurry flowability and backfill strength are key for safe and economic mining [9,10]. The flowability of the slurry and the strength and macro-pore structure of the backfill are related. Therefore, it is necessary to analyse the correlation between the porous structure of the backfill and the macro parameters. To study the porous structure of CPB, most studies use scanning electron microscopy (SEM) for microstructure analysis and nuclear magnetic resonance (NMR) for porosity analysis. Koohestani et al. [11] used SEM images to qualitatively analyse the structure and composition of wood-pulp-filled pores. Based on the NMR and mercury intrusion experiments of the porous structure and components, Fridjonsson et al. [12] explained the macro parameters, including uniaxial compressive strength (UCS) and permeability. Ai [13] used NMR to analyse changes in porous structure. Hou et al. [14] established the relationship between NMR porosity and slurry concentration using NMR experiments. Li et al. [15] used SEM to identify and analyse the hydration reaction products of backfill, such as AFt. In general, the research and analysis of SEM images are mainly based on qualitative analysis. The research contents include rough aperture measurement, hydration product identification, and composition and structural analysis. Zhang et al. [16] and Saper [17] introduced the stereological principle as the theoretical basis for the quantitative analysis of material cross sections. The stereological principle is the theoretical basis for establishing the correlation between the research object and its cross section. Based on the principle of stereology, the three-dimensional results can be determined using a two-dimensional section of the material. This theory is widely used in the study of material cross sections to verify the data obtained from SEM and CT images. Additionally, the properties of the research object are obtained. The consistency of the two-dimensional image of the material can be added to the quantitative analysis [18]. The fractal theory proposed by Mandelbrot [19] is introduced as a research method to enrich the structural analysis of the research object. In the past two years, Tian et al. [20] introduced fractal dimension calculations in the CT scanning of concrete freeze–thaw damage. Wang et al. [21] calculated the fractal dimension of natural diatomite SEM images. The fractal dimension [22] is used as the characterization method for the surface profile of corroded steel bars. The fractal dimension is also applied to study human bone structure and natural material structure [23,24]. In general, when using fractal dimensions to analyse the porous structure of materials, a porous structure with a higher disorder is considered to show a higher fractal dimension. The quantitative analysis of the porous structure of backfill is used as the basis of theory and application. Furthermore, the fractal dimension can be correlated with other experimental parameters. Sun et al. [25] established the relationship between the fractal dimension and methane adsorption by analysing the fractal dimension of SEM images of marine shale. Many other researchers [26–29] introduced fractal dimensions to analyse the correlations between structural fractal dimensions and other parameters in geotechnical studies. In these studies, fractal dimension data of research objects were mostly obtained by mercury intrusion experiments, CT scans and SEM images. For SEM images, the fractal dimension is obtained through a series of operations for image recognition, processing, calculation, etc. [30–32]. The experimental results show that the porous structure of the rock mass and cemented materials has obvious fractal characteristics on a microscopic scale [33,34].

The fractal theory improves the degree of data mining of SEM images. Based on the theory, quantitative analysis of the SEM images of the microscopic porous structure can promote the correlation between SEM and

**Table 1.** The basic physical properties of stone powder and tailings.

| raw materials | apparent density (g cm$^{-3}$) | packing density (g cm$^{-3}$) | surface moisture content (%) |
|---|---|---|---|
| Tailings A | 3.49 | 1.24 | 0.128 |
| Tailings B | 2.77 | 1.17 | 0.135 |
| Stone powder | 2.89 | 0.99 | 0.162 |

NMR experimental results. However, the analysis of the SEM images of the backfill porous structure mostly stays at the qualitative level. Little research has been conducted on the correlation of multiscale parameters between the microstructure and macro parameters of backfill. Based on the theory of stereology and fractal dimension, the stone powder cement tailings backfill (SPCTB) is taken as the research object in this work. The images were processed, including image contrast adjustment, image noise reduction, grey scaling and binarization. Following this processing, the fractal dimensions of the SEM images of the microscopic porous structures were calculated using OTSU and box dimension methods. The quantitative correlations, which were between the fractal dimension and macro parameters of the filling slurry, were calculated in terms of equilibrium shear stress (ESS) and equilibrium apparent viscosity (EAV). In addition, the quantitative correlations between the fractal dimensions and macro parameters of backfill in terms of UCS, water content (WC) and porosity were obtained. The relevance between the fractal dimension of the micropore structure of backfill and macroscopic experiment results was established, providing a new approach for quantitative studies across scales.

# 2. Materials and methods

## 2.1. Raw materials

The tailings samples used in the study were obtained from the Gaofeng mine in Guangxi Province, China. The collected samples were divided into two types: Tailings A and Tailings B. Tailings A is produced by an old-fashioned beneficiation process, and Tailings B is produced by a new type of beneficiation in a tin ore. The siliceous limestone was obtained from the quarry around the Gaofeng mine. In this experiment, the mine waste siliceous limestone samples were ground in a horizontal ball mill with a volume of 35 l; the cylinder's rotational speed was 150 r min$^{-1}$, and the processing time was 20 min. The limestone, which was used to replace part of the cement, was made into stone powder with a certain particle size range. The basic physical properties of the stone powder and tailings are shown in table 1. P42.5 cement with a strength grade of C30, produced by the Changsha Xinxing Cement Factory, was selected as the cementing material. The main elements of the experimental raw materials were analysed using Dutch PANalytical X-ray fluorescence, and the results are presented in table 2. Additionally, Mastersizer 2000 was used to analyse the particle size of the raw materials, and the results are shown in figure 1. The SEM images of the stone powder and tailings are shown in figure 2.

## 2.2. Slurry mix

The slurry has a mass concentration of 70% and a lime-to-sand ratio of 1 : 4. The control variable is the amount of stone powder used to partly replace the cement. Groups A1 and B1 without the stone powder are set as the control groups. Three samples are taken from each group to reduce the error. The size of each sample is 70.7 mm$^3$. The size of each sample for NMR tests is 15 mm$^3$. The tailings and stone powder are regarded as an aggregate. The unevenness coefficient Cu $= d60/d10$ and the curvature coefficient Cc $= (d30 \times d30)/(d60 \times d10)$ are introduced. Generally, aggregates with Cu $> 10$ and $1 < $ Cc $< 3$ are considered to have good grading. However, if Cu is too large, the particle size needs to be adjusted. Furthermore, $d_p$ represents the particle size value of the cumulative proportion of $p$% and is obtained using equation (2.1):

$$d_p = d_1 + \frac{(p - p_1)(d_2 - d_1)}{p_2 - p_1},$$ (2.1)

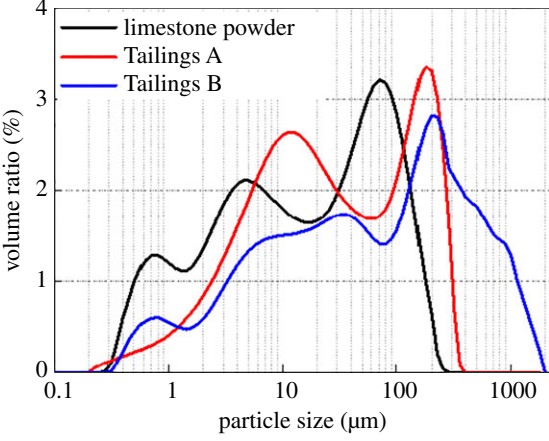

**Figure 1.** Particle size distribution of the raw materials.

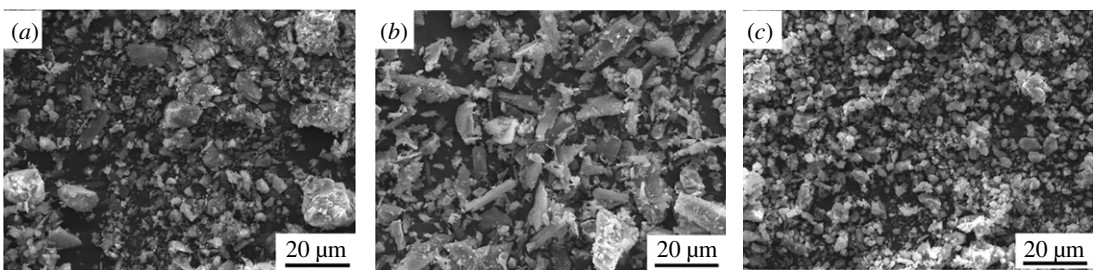

**Figure 2.** SEM images of the stone powder and tailings; figures (*a*), (*b*) and (*c*) represent Tailings A, Tailings B and stone powder, respectively.

**Table 2.** Primary elemental concentrations of raw materials.

| element | Tailings A (%) | Tailings B (%) | limestone powder (%) |
|---|---|---|---|
| Mg | 2.12 | 0.40 | 0.58 |
| Al | 1.11 | 3.06 | 0.69 |
| Si | 4.48 | 16.24 | 14.20 |
| S | 15.86 | 8.88 | 0.24 |
| K | 0.50 | 0.67 | 0.18 |
| Ca | 14.03 | 9.09 | 33.93 |
| Mn | 0.14 | 0.41 | — |
| Fe | 23.10 | 12.66 | 0.67 |
| Zn | 1.23 | 1.14 | — |
| As | 0.99 | 1.48 | — |
| Sn | 0.37 | 0.24 | — |
| Sb | 0.32 | 0.35 | — |
| Pb | 0.34 | 0.52 | — |

where $d_p$ is the particle size to be determined, $p$ is the target ratio, $p_1$, and $p_2$ are the known adjacent ratios of $p$ and $d_1$ and $d_2$ are the particle sizes corresponding to the known adjacent ratios.

The unevenness coefficient Cu and the curvature coefficient Cc of each group are calculated, and the proportion of each component in the slurry and the corresponding evaluation parameters are presented in table 3.

**Table 3.** The mixture ratio of the slurry and grading parameters of the mixed aggregate.

| group | mixture ratio | | | | grading parameters | |
|---|---|---|---|---|---|---|
| | tailings (%) | water (%) | limestone powder (%) | cement (%) | Cu | Cc |
| A1 | 56 | 30 | 0 | 14.0 | 14.56 | 0.72 |
| A2 | | | 1.4 | 12.6 | 14.68 | 0.72 |
| A3 | | | 2.1 | 11.9 | 14.80 | 0.73 |
| A4 | | | 2.8 | 11.2 | 14.93 | 0.73 |
| B1 | | | 0.0 | 14 | 40.02 | 0.71 |
| B2 | | | 1.4 | 12.6 | 44.36 | 0.64 |
| B3 | | | 2.1 | 11.9 | 44.10 | 0.64 |
| B4 | | | 2.8 | 11.2 | 43.83 | 0.65 |

## 2.3. Methodology

Two kinds of process tailings and waste limestone powder were obtained from local mines and used as aggregates. Eight sets of SPCTB samples were obtained with different proportions and curing times. The experiments explored the correlations between the fractal dimension of the porous structure and the macro parameters of the backfill and filling slurry. The ESS and EAV of the filling slurry were tested using a HAAKE VT550 rotational rheometer. The UCS was evaluated using a uniaxial compression apparatus. The WC and porosity were measured using the Ani-MR150 rock magnetic resonance imaging analysis system. The SEM analysis of the porous structure was performed using a Czech TESCAN MIRA3 field emission scanning electron microscope. The MATLAB-FRACLAB toolbox was used to transform the SEM images into greyscale, calculate the image-threshold and binary images and then calculate the fractal dimension. The micropore fractal dimension data of each sample were obtained. The correlation analysis was conducted between the fractal dimension and macro parameters in terms of shear stress and apparent viscosity strength to explore the strength of correlation for each parameter.

# 3. Results

## 3.1. Flowability of the filling slurry

From the results of eight sets of constant shear tests (shear rate of $10\,s^{-1}$), the change in shear stress is related to time. The results are shown in figure 2b and figure 3a. The apparent viscosity is related to time, as shown in figure 2d and figure 3c.

It can be seen from figure 3 that, under different tailings properties and amounts of stone powder, the slurry exhibits obvious time-varying characteristics of shear thinning. The shear stress and apparent viscosity gradually increase with time. After a certain value, these parameters decrease and gradually stabilize. Considering a mixing ratio of limestone powder of 1.4%, it is observed that, when the shearing time is 0 s, the shear stress of Slurry B (including Tailings B) is 227.22 Pa larger than that of Slurry A (including Tailings A). The apparent viscosity of Slurry B is 408.74 Pa · s larger than that of Slurry A. When the equilibrium state is reached, the shear stress of Slurry B is 107.12 Pa larger than that of Slurry A, whereas the apparent viscosity of Slurry B is larger than that of Slurry A by 303.46. Pa · s. Similar characteristics are observed for mixtures containing other contents of stone powders. It can be seen that the flocculant network structure formed by Tailings B is more stable and requires more force to destroy. In addition, when the mixing ratio of limestone powder is 0, after 211 s, the shear stress of Slurry A reaches an equilibrium value of 81.14 Pa. When the mixing ratio of limestone powder increases to 2.1%, it takes 313 s for Slurry A to reach the equilibrium value of 75.87 Pa. When the mixing ratio of limestone powder is 0 and 2.1%, the apparent viscosity of Slurry A reaches equilibrium values of 134.41 Pa · s and 110.43 Pa · s, respectively.

The equilibrium time for Slurry B for different proportions of stone powder is approximately the same as that of Slurry A. The larger the amount of stone powder, the longer it takes the slurry to

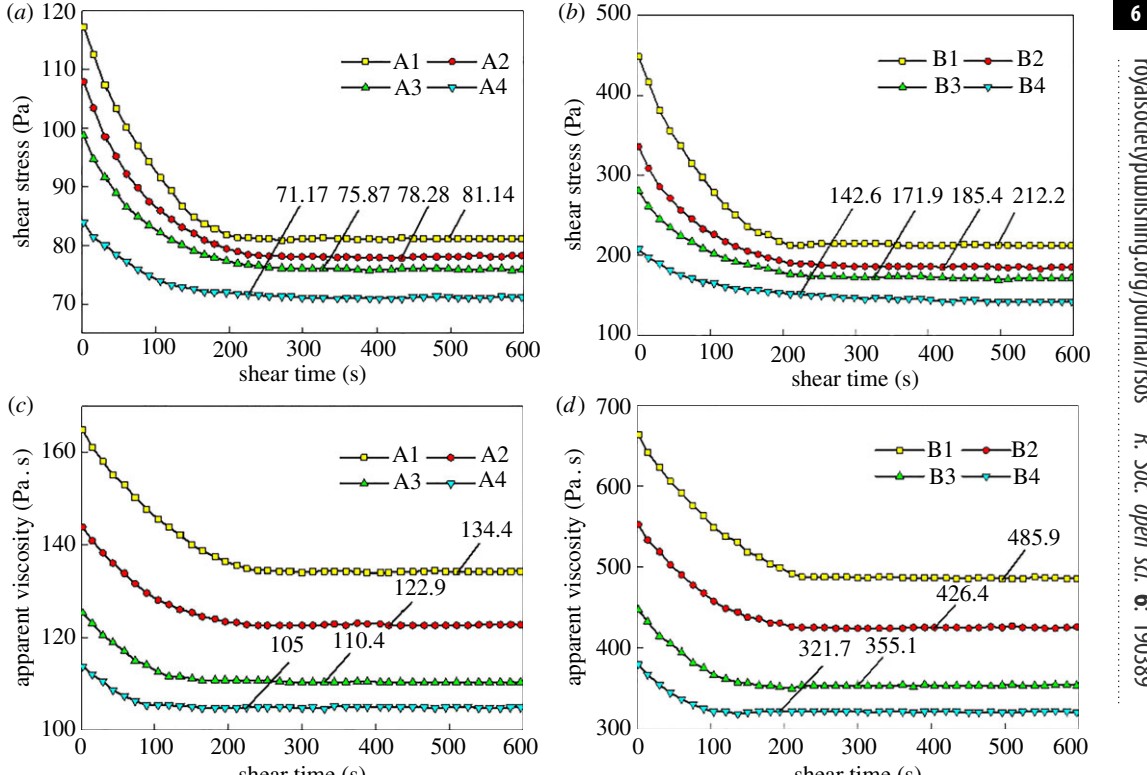

**Figure 3.** Flowability of the filling slurry: (*a,b*) are the shear stress for Groups (A) and (B) and (*c,d*) are the apparent viscosities for Groups (A) and (B).

reach equilibrium and the smaller the values of ESS and equilibrium viscosity. The reason is that the fracturing structure formed by stone powder and slurry is easy to break and destabilize. When the amount of stone powder is increased, the internal flocculant network structure is easily destroyed and the time-varying characteristics of shear thinning are more obvious. The flocculant network structure hinders the pipeline transportation of slurry. Consequently, a small amount of stone powder can be added before the transportation of slurry to form an unstable flocculant network structure to reduce the dynamic viscosity of the slurry and reduce energy loss during transportation. This phenomenon, in turn, enhances the slurry conveying performance. However, the unstable flocculant network structure has a certain impact on the cement hydration reaction in the later stage, which reduces the strength of the backfill.

## 3.2. Strength and porosity experiments

Porosity theory concerns the science of pores and states that the effect of the pore structure on the macroscopic behaviour of concrete is more important than that of porosity [35–37]. For the same pore volume and distribution of pores in different shapes and quantities, the fractal dimensions differ. The fractal dimension of the pore structure is used as a quantitative tool for describing the distribution of pores. The fractal dimension of the microscopic porous structure of backfill can objectively reflect the pros and cons of the structure. For CPB, factors such as aggregate gradation, aggregate composition, curing time and product structure affect the fractal dimension. The relevant data obtained from the UCS and NMR experiments are presented in table 4. The strength data are selected at 7 and 28 days. WC was tested after 7 days because the samples cannot maintain water, while NMR porosity was obtained after 28 days using NMR technology.

## 3.3. SEM image preprocessing

Image preprocessing includes two main processes: image contrast adjustment and noise reduction.

**Table 4.** Macro-experimental results of SPCTB, including UCS, WC and porosity.

| group | UCS test | | NMR experiments | |
| --- | --- | --- | --- | --- |
| | 7 days strength (MPa) | 28 days strength (MPa) | 7 days WC (%) | 28 days porosity (%) |
| A1 | 0.985 | 1.652 | 0.71 | 1.99 |
| A2 | 0.838 | 1.565 | 1.69 | 2.39 |
| A3 | 0.740 | 1.555 | 1.81 | 2.69 |
| A4 | 0.601 | 1.575 | 1.92 | 2.51 |
| B1 | 0.715 | 1.530 | 13.20 | 14.13 |
| B2 | 0.661 | 1.245 | 6.47 | 13.41 |
| B3 | 0.588 | 1.125 | 7.16 | 13.37 |
| B4 | 0.535 | 1.390 | 10.87 | 13.12 |

### 3.3.1. Image contrast adjustment

Contrast adjustment is the basis of image edge extraction and image segmentation. By adjusting the image grey histogram, the colour difference between the target area and the background area can be enhanced. Consequently, target features can be enhanced, and background features can be suppressed. This study used *decorrstretch* (de-correlation stretching), *adapthisteq* (finite contrast adaptive histogram equalization), *histeq* (histogram equalization) and *imadjust*, *stretchlm* and *imcontrast* (contrast adjustment). Finally, the *imcontrast* function was used to adjust the image contrast, i.e. an image contrast adjustment dialog box is used to realize the real-time observation adjustment effect and directly extract the optimal adjustment results.

### 3.3.2. Image noise reduction

Image noise can cause image blurring, pitting, etc., which directly affects the image binarization results and causes errors in the quantitative analysis of images. In this study, the effect of noise on quantitative analysis is reduced by image noise reduction operations and morphological operations after image segmentation. This study tested and compared *medfilt2* (two-dimensional median filtering), *ordfilt* (two-dimensional sorting statistical filtering), *wiener2* (two-dimensional Wiener filtering), *filter2* (mean filtering), *DTC* (discrete cosine variation), *deconvblind* (blind deconvolution operation) and a total of six methods for noise reduction. To preserve the image details as much as possible during noise reduction, the *filter2* mean filtering method (average mean filter) was used to reduce image noise. The noise reduction operation causes the tailings particle area to blur. Therefore, while denoising SEM images of different quality backfills, the parameters should be carefully selected.

## 3.4. SEM image segmentation

### 3.4.1. Greyscaling of SEM images

Using MATLAB software, SEM images of backfill samples with various ages and proportions were converted to greyscale images. The appropriate binarized greyscale segmentation thresholds were calculated, and the greyscale images were further binarized. The original SEM images had a total of $1024 \times 760 = 778\,240$ pixels. Their colour was similar to that of the greyscale image. According to the MATLAB function, the original SEM images were still coloured images. Consequently, the original SEM images needed to be converted to greyscale images through greyscale conversion so that greyscale threshold calculations could be performed. The grey value was divided into a 256-step greyscale (0–255). The floating point algorithm was used for the original SEM images. The grey value of the images was calculated using the expression $\text{Grey} = R \times 0.3 + G \times 0.59 + B \times 0.11$. Three primary colours can be sequentially converted to greyscale based on the images of a single pixel point colour. The grey level of the corresponding pixel could be obtained. The SEM images were converted to greyscale images by the rgb2gray function of MATLAB. The results are shown in figure 4.

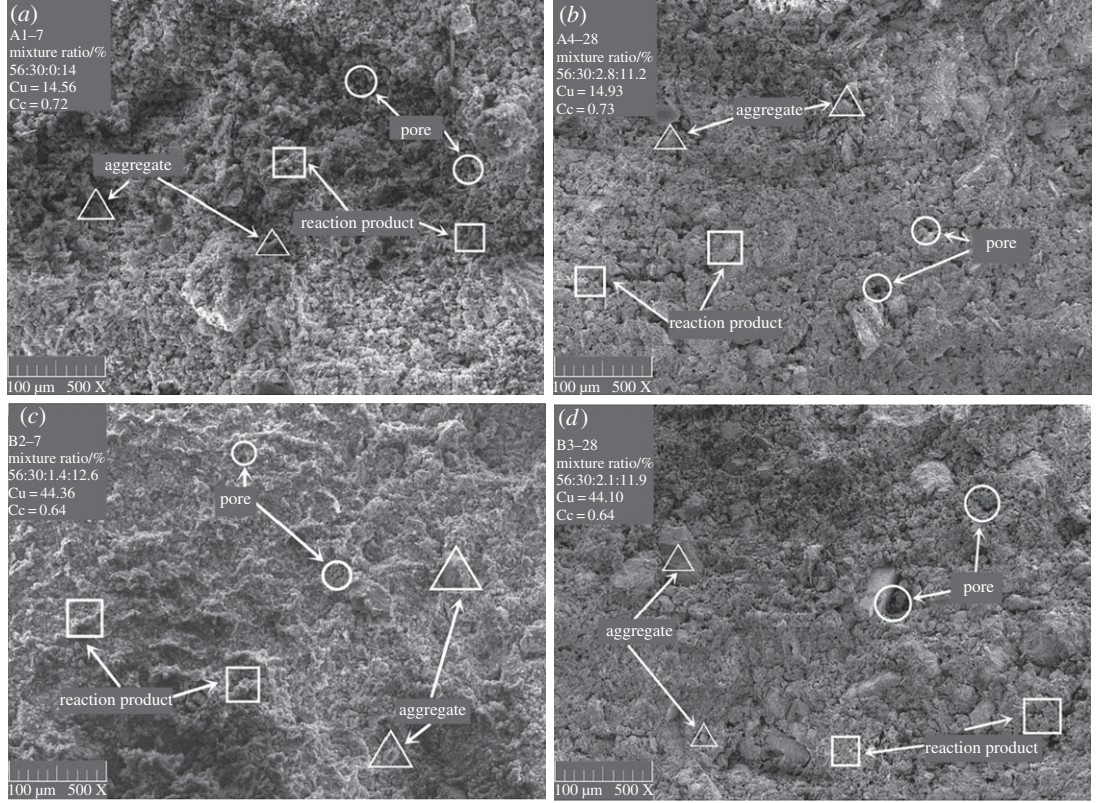

**Figure 4.** ($a$–$d$) SEM greyscale images of SPCTB after MATLAB processing.

### 3.4.2. Greyscale binarization

OTSU is an adaptive method for determining the image segmentation threshold. The overall process is shown in figure 5. According to the greyscale characteristics of an image, the method divides the image into background and target (for the SEM image of backfill, the more porous structure is defined as the background, while the reaction product and the aggregate with low greyscale are defined as the targets). By traversing the pixel points of greyscale images and performing the determination, the variance between the pixels and the average greyscale of the whole image is calculated. Taking the increase in variance as a criterion, when the variance is maximized, the division is optimal. This algorithm can minimize the probability of misclassification [38]. In MATLAB, OTSU can be called by the function to calculate the image segmentation threshold. By calling the *graythresh* function, the 256-step greyscale is equally divided into proportional values in [0, 1]. The segmentation threshold is calculated for SEM greyscale images, and the results are presented in table 5.

Based on the greyscale segmentation threshold, the individual pixels of the greyscale images are binarized one by one. The binarization judgement function is given by equation (3.1):

$$f(i, j) = \begin{cases} 0 & f(i, j) \leq T \\ 1 & f(i, j) > T' \end{cases} \tag{3.1}$$

where $T$ is the segmentation threshold.

The pixel of grey levels lower than or equal to the segmentation threshold $T$ is set to 0, which indicates a black pixel point, by the 2bw binarization function in MATLAB. A pixel point whose grey level is higher than $T$ is set to 1, representing a white pixel point. The SEM binarized images and their comparison with the greyscale images are shown in figure 6.

### 3.5. Calculation of the fractal dimension

For practical problems, if the research object has no obvious self-similarity, the box dimension method is used to calculate the fractal dimension. The box dimension of SEM images of backfill is tested using the FRACLAB toolbox for fractal dimension calculation in MATLAB (when the SEM image is complex, other

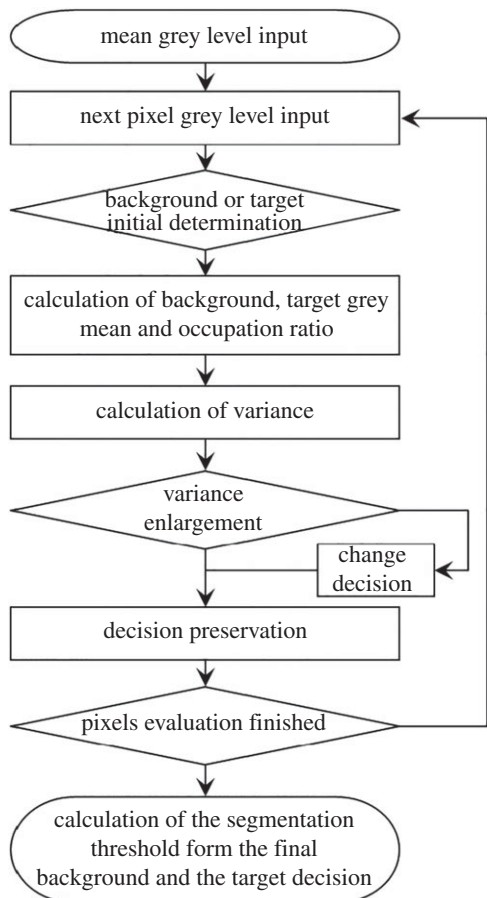

**Figure 5.** The process for calculating the grey threshold of OTSU.

**Table 5.** The calculation results of the grey threshold at two ages.

| group | 7 days of Group A | 28 days of Group A | 7 days of Group B | 28 days of Group B |
|---|---|---|---|---|
| 1 | 0.41 | 0.49 | 0.43 | 0.46 |
| 2 | 0.40 | 0.59 | 0.45 | 0.38 |
| 3 | 0.43 | 0.57 | 0.43 | 0.38 |
| 4 | 0.42 | 0.40 | 0.43 | 0.49 |

software tools are needed for the calculation. The local sampling magnified image of the A1–7 day sample is used for the principle of figuration). The box dimension calculation process of A1–7 day samples of the SEM partial image is shown in figure 7.

By setting the aspect ratio and $X$, the toolbox equally divides the image into $[0–X]$ times. After the division, whether there is a background area in the unit aliquot area is determined (that is, there is a pixel value of 0). The equally divided area of the background is recorded as a 'box'. By multiple division and counting steps (figure 7$a$), a $2^X$–$2^Y$ double-index coordinate graph consisting of an equal number of boxes is obtained (figure 7$b$). The slope of the fitted line represents the fractal dimension [21,39].

Some of the SEM image box dimensions are shown in figure 8, where $X$ is the number of equal divisions, $2^{-X}$ is the relative side length of the unit box with respect to the SEM images and $2^Y$ is the number of boxes. It should be noted that, when $X$ exceeds a certain value, the number of boxes tends to remain the same due to the limitation of image resolution. For the SEM images used in this experiment, the upper limit of boxes is $2^{19.57}$. When $X > 10$ is selected, the value of $N$ tends to be constant, and therefore, $X = [0–10]$ is limited. In this equal division scale, the results show significant fractal features. The test points are linearly fitted based on the least-squares method, and the linear slope represents the box dimension of the SEM images.

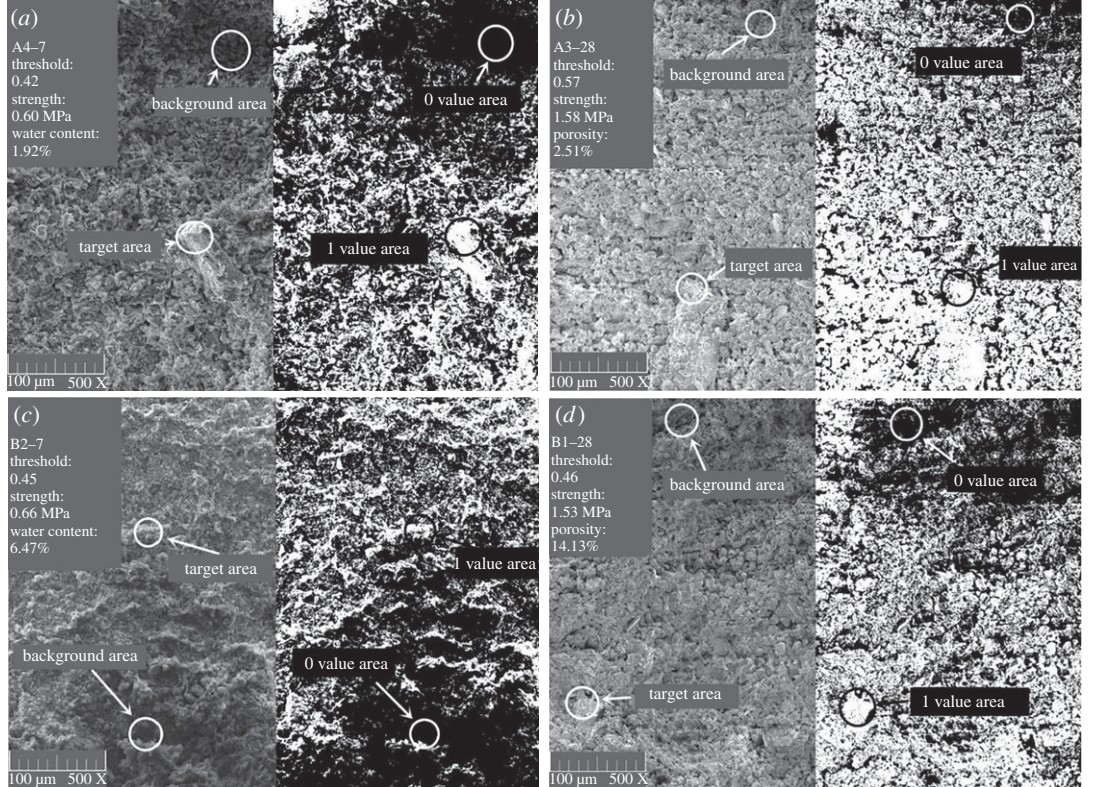

**Figure 6.** (a–b) Background and target area in greyscale images corresponding to the 0 and 1 value areas in binarized images.

By obtaining the slope of the fitted line, the corresponding box dimension of the SEM microscopic porous structure of the images and the box dimension variation of the 28-day group with respect to the 7-day group were obtained, as shown in figure 9.

# 4. Discussion

## 4.1. Fractal dimension characteristics

The average fractal dimensions of Groups A-7, B-7, A-28 and B-28 (figure 9a) are 1.789, 1.790, 1.802 and 1.803, respectively. After a longer period of maintenance, the fractal dimensions of the porous structure, formed by the backfill including different tailings components, have improved. At the same time, under the same tailings components and different ratios, the fractal dimension of the backfill also improved (figure 9b). This result indicates that the porous structure becomes disordered due to the hydration reaction, and the structural complexity increases, mainly because the C–H–S gel structure, formed by the hydration reaction, is complicated. As the hydration reaction proceeds, the proportion of C–H–S gel increases, increasing the fractal dimension of the porous structure.

The fractal dimension of the two backfill groups at 28 days (figure 9a) showed a higher consistency than the experimental results at 7 days. Among the samples aged 28 days, the control group without the stone powder had the largest difference. The consistency of the groups with added stone powder increased, indicating that the fractal dimension of the porous structure tends to stabilize with the continuation of the hydration reaction. Additionally, the fractal dimension of the porous structure can be affected by the change in the mixing ratio.

At the same age, the fractal dimension of Group B is only 0.06% units higher than that of Group A (figure 9a), indicating that the porous structures of Groups A and B are quite complex. The grading and particle size of Groups A and B are analysed (table 3). The aggregate unevenness coefficient $C_u$ of Group B is much higher than 10, indicating that there is a serious loss of intermediate particle size. The curvature coefficient $C_c$ of Group B is inferior to that of Group A, indicating that, theoretically, the porosity of Group B is greater than that of Group A and that the distribution of pores is inferior to that of Group A, which is consistent with the experimental data for porosity (table 3). Further analysis

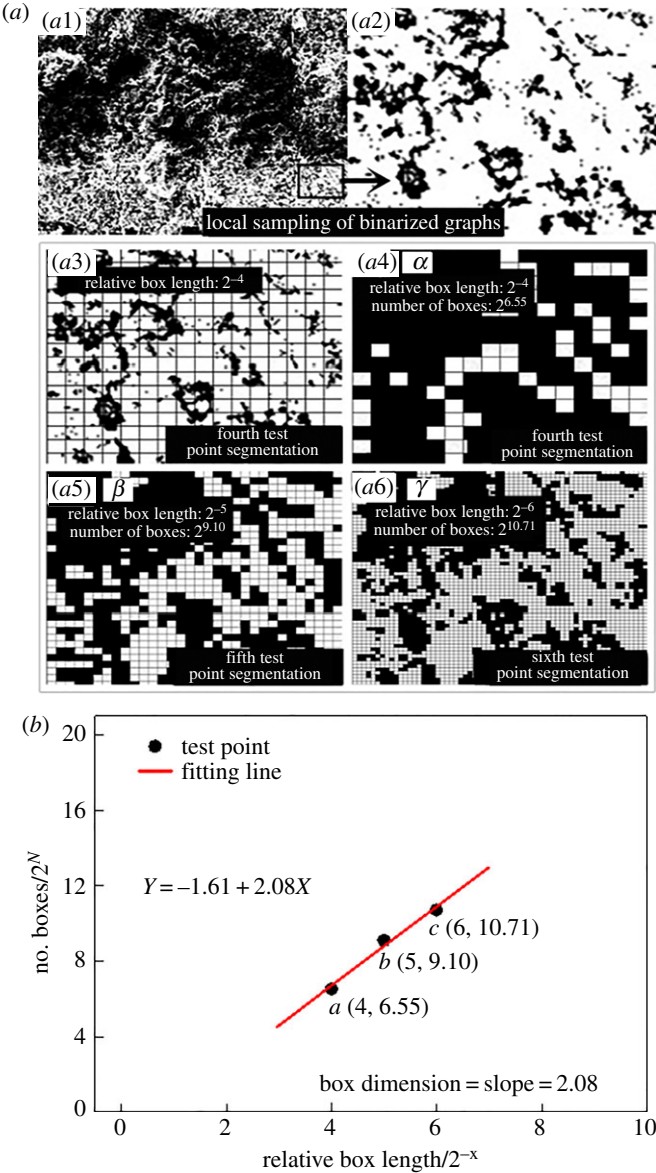

**Figure 7.** Two steps of box dimension calculation: (*a*) one example of box dimension test and (*b*) linear fitting of test results.

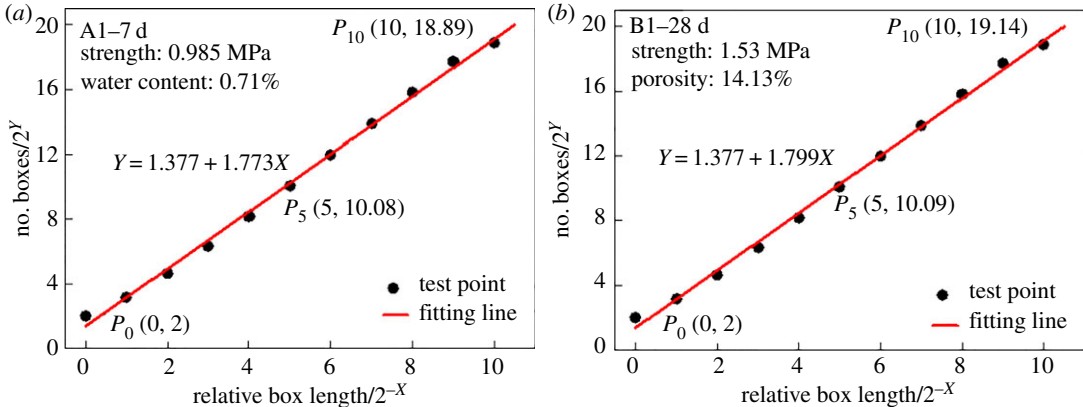

**Figure 8.** (*a,b*) Box dimension test and linear fitting of Groups A1−7d and B1−28d.

shows that, with the appearance of the C−H−S gel structure, the influence of Cu and Cc is reduced to some extent. In the experiments, the influence of gradation and porosity on the porous structure is unclear because the porous structure cannot be inferred from the gradation and porosity of

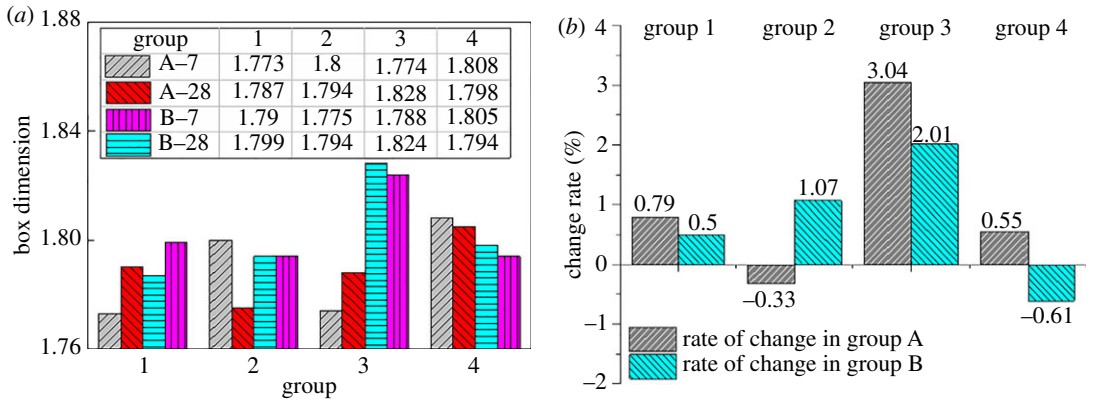

**Figure 9.** Box dimension analysis of the porous structure of the SEM images: (a) box dimension comparison and (b) rate of change of the box dimension.

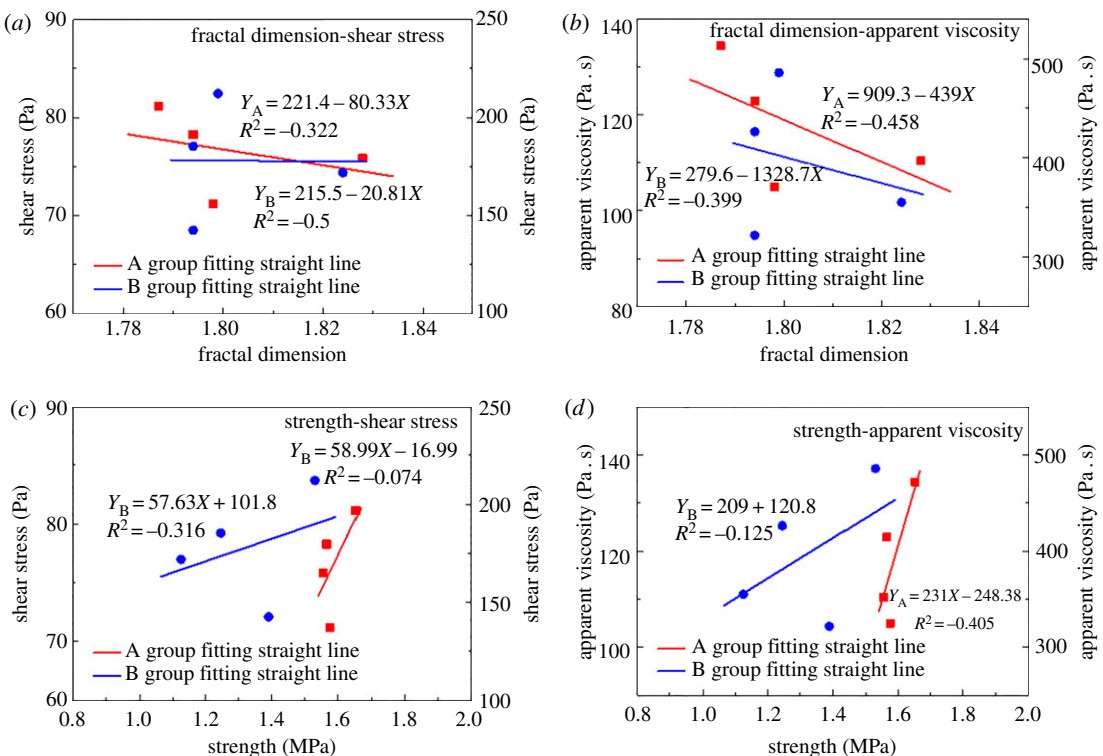

**Figure 10.** (a–d) Trans-scale correlation calculation of the parameters for flowability.

aggregates. When performing the pore-related parametric analysis, the porous structure should be discussed separately from certain parameters such as porosity, pore distribution and gradation [40].

## 4.2. Correlation analysis of the flowability macro parameters of the filling slurry

The flowability of the filling slurry is related to the pipeline's wear during the transportation of the slurry. Various properties of the filling slurry are also closely related to the strength and porous structure of the backfill after the hardening of the filling slurry. The correlation between the three parameters is analysed. Pearson product-moment correlation coefficient analysis is conducted for the flowability of filling slurry in terms of ESS and EAV, and the mechanical properties of the backfill are studied in terms of UCS and porous fractal dimension. The results of the mean square error calculation and the results for linear fitting are shown in figure 10.

The ESS of the slurry shows a strong correlation with the pores' fractal dimension, and the absolute value of the variance $|R^2|$ exceeds 0.3 (figure 10a). The ESS and EAV of Groups A and B are negatively

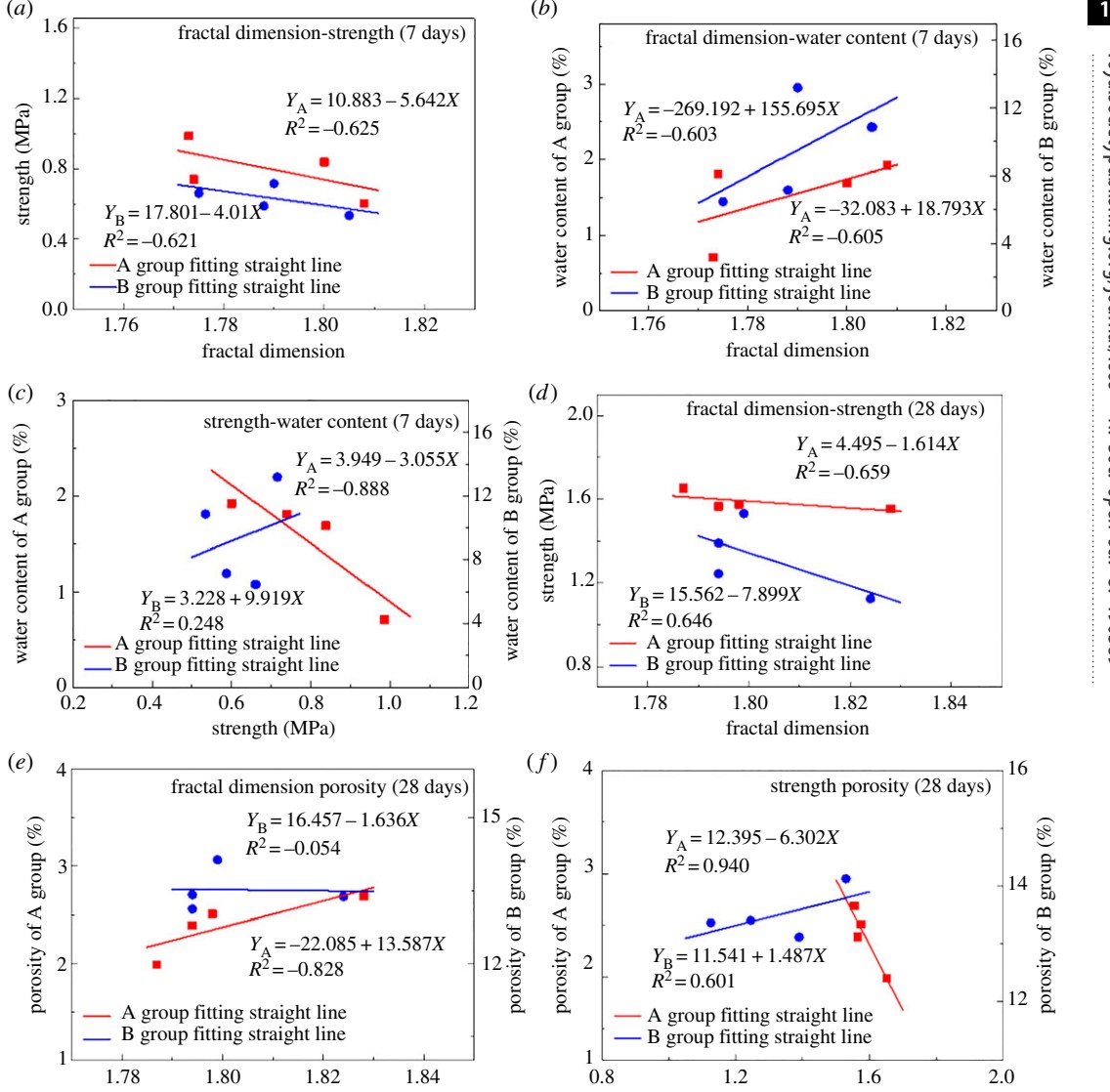

**Figure 11.** ($a$–$f$) Trans-scale correlation calculation of the parameters for mechanical properties.

correlated with the pores' fractal dimension (figure 10$a$,$b$), which means that smaller values of EAV are associated with better flowability of the filling slurry. Consequently, the pipelines consume more energy, which results in a higher pore fractal dimension of hardened filling slurry and a more complex porous structure. This result occurs because, when more stone powder is added to the slurry, the flocculant network structure formed inside the slurry becomes unstable. The structure is easy to break, which reduces energy consumption during pipeline transportation. However, the flocculant network structure is easy to destroy, and the stability of the structure's inner backfill is poor. The flocculant network structure during the hydration period exhibits poor stability, and therefore, there is a more broken flocculant network structure and the porous structure also becomes complex.

The correlations among strength, ESS and EAV are generic in nature. The absolute value of the correlation coefficient between the strength and ESS of Group B is 0.316. The absolute value of the correlation coefficient between the strength and EAV of Group A is 0.405, indicating that the strength has a certain positive correlation with ESS and EAV. When the ESS and EAV of the filling slurry are large, the strength of the backfill correspondingly increases. This result shows that, although some energy is consumed during pipeline transportation, the quality of backfill formed in the later stage is better. Therefore, it is necessary to find an optimum point to reduce the energy consumed at the time of pipeline transportation and ensure the strength of backfill. The measure adds an appropriate amount of stone powder into the slurry to reduce and adjust the ESS and EAV values.

## 4.3. Correlation analysis of the mechanical properties of the macro parameters of backfill

The Pearson product-moment correlation coefficient is used to analyse the experimental data, such as the UCS, WC, porosity and fractal dimension of backfill. The results of mean square error calculation and linear fitting are shown in figure 11.

The fractal dimension of the porous structure shows a strong correlation with the backfill strength. The absolute value of the variance $|R^2|$ exceeds 0.6. The Pearson correlation coefficient at 28 days between the fractal dimension and strength is higher than that at 7 days, which indicates that the correlation between the fractal dimension and strength has increased with the increase in curing time.

The correlation between the fractal dimension and strength at 7 days (figure 11$a-c$) shows a strong negative correlation, indicating that an increase in the strength reduces the fractal dimension. The strength of the backfill is affected by the complexity of the porous structure. The positive correlation between the fractal dimension and WC demonstrates that the increase in the fractal dimension will increase the WC, and the change in the complexity of the porous structure changes the surface area of the pores. The change in the surface area of the pores affects their water storage capacity. The correlation between strength and WC is not clear.

The fractal dimension-strength correlation at 28 days maintains a strong correlation (figure 11$d$), and the correlations among porosity, fractal dimension and strength are unstable (figure 11$e$). For Group A with lower porosity (table 4), the porosity is strongly correlated with the other two parameters. For Group B with higher porosity, the correlations between dimension and porosity and between strength and porosity are unclear because the aggregate gradation of Group B (figure 1 and table 3) is missing the intermediate grain size. There is a certain amount of large particle tailings, which cause the value of Cu to be too large and adversely affect the backfill parameters. For CPB with high porosity and gradation, the parametric correlation needs to be further studied.

## 5. Conclusion

   (i) Using the box dimension analysis method, the fractal dimension of the SPCTB's porous structure is obtained. The value varies in the range from 1.773 to 1.828, and the average value is 1.796. The fractal dimension value indicates that SPCTB has a relatively stable porous structure, which helps maintain the stability of the backfill.
  (ii) The fractal dimension of the backfill mainly characterizes the complexity of the porous structure. In the time span, the fractal dimension difference from 7 to 28 days shows that the fractal dimension of the porous structure increases with increasing curing time. The main reason for this result is that the generated C–H–S gel structure increases the complexity of the backfill microstructure due to the influence of the hydration reaction of the cementitious material.
 (iii) A trans-scale relationship function between the fractal dimension of the SEM microporous structure of backfill and the ESS and EAV of the filling slurry parameters is established. There is a negative correlation between the fractal dimension of the porous structure and the parameters in the filling slurry. The UCS of the backfill is positively correlated with the macro parameters of the filling slurry.
 (iv) The trans-scale relationship function between the fractal dimension of the SEM microporous structure and the UCS and porosity of the backfill is established. Under certain conditions, the fractal dimension has a negative correlation with the backfill strength. The mean square error is $R^2 = -0.638$. The fractal dimension is positively correlated with WC, and the mean square error is $R^2 = 0.604$.

Ethics. We have obtained publication permission from all authors, and we declare that the present experiments and manuscript were performed in accordance with the standard of academic conduct from Chinese academic societies. No experiments in this study included human studies or field studies on animals.

Data accessibility. The datasets supporting this article have been uploaded as part of the electronic supplementary material.

Authors' contributions. J.h.H. and Q.f.R. designed the experimental process. Q.f.R. and Q.J. prepared the stone powder cement tailings backfill samples. Q.J. and X.t.D. tested the strength of all samples and finished other tests. Q.f.R. and Q.J. collected and analysed the data. J.h.H. and Q.f.R. interpreted the results and wrote the manuscript. All authors gave final approval for publication.

Competing interests. The authors declare that they have no competing interests.

Funding. This research was supported by the National Key Research and Development Program of China (2017YFC0602901) and the National Natural Science Fund (41672298) of the Ministry of Science and Technology of the People's Republic of China.

Acknowledgements. The authors thank the management and staff of Gaofeng Mine for their valuable support, the instructional support specialist of Modern Analysis and Testing Central of Central South University and the two anonymous reviewers for their helpful comments.

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
