## [Reviewer comments · Royal Society Open Science]

Review History

RSOS-190389.R0 (Original submission)

Review form: Reviewer 1

Is the manuscript scientifically sound in its present form?

Yes

Are the interpretations and conclusions justified by the results?

Yes

Is the language acceptable?

Yes

Is it clear how to access all supporting data?

Yes

Do you have any ethical concerns with this paper?

No

Have you any concerns about statistical analyses in this paper?

No

Recommendation?

Accept with minor revision (please list in comments)

Comments to the Author(s)

The research was carried out in a comprehensive way which gives sufficient credibility to the main results. The particular value of the paper is in its original approach for analysing the results. Interesting paper with an interesting approach that few researchers considered previously in this field. However, I would like to recommend the following improvements:

1. In the Matlab analysis for the obtained SEM micro-images, authors did not mention the dilemma of noise in the background. How authors managed to reduce the noise in the images for a proper analysis?

2. In the abstract, authors have mentioned the OTSU method. In which part of the research did authors use this method? I could only notice that the authors have used the box-counting method.

3. In section 2.1 (Raw materials), please provide more information about Tailings A and B.

4. The introduction part is very short and needs to be extended. Despite the presence of very few studies that discuss such materials and even the techniques that were used in this research, authors need to explain more about other materials that were used to enhance the performance of cement. Following papers could help:

- A novel approach of introducing crystalline protection material and curing agent in fresh concrete for enhancing hydrophobicity.
- Development of hydrophobic concrete by adding dual-crystalline admixture at mixing stage.
- Influence of early water exposure on modified cementitious coating.

The paper is very interesting and I recommend it for publication but after the authors rectify the comments.

Review form: Reviewer 2**Is the manuscript scientifically sound in its present form?**

Yes

Are the interpretations and conclusions justified by the results?

Yes

Is the language acceptable?

No

Is it clear how to access all supporting data?

No

Do you have any ethical concerns with this paper?

No

Have you any concerns about statistical analyses in this paper?

No

Recommendation?

Accept with minor revision (please list in comments)

Comments to the Author(s)

The relation between micro porous structure and macro parameters of backfills is difficult to be quantitatively calculated. This paper shares a method through calculating the fractal dimension of pores. It is really an interesting topic.

The major problem is the language. The English needs substantial editing. So I recommend that this paper can be published after language polish.

Decision letter (RSOS-190389.R0)

01-Apr-2019

Dear Dr Ren,

The editors assigned to your paper ("Trans-scale relationship analysis between pore structure and macroparameters of backfill and slurry") have now received comments from reviewers. We would like you to revise your paper in accordance with the referee and Associate Editor suggestions which can be found below (not including confidential reports to the Editor). Please note this decision does not guarantee eventual acceptance.

Please submit a copy of your revised paper before 24-Apr-2019. Please note that the revision deadline will expire at 00.00am on this date. If we do not hear from you within this time then it will be assumed that the paper has been withdrawn. In exceptional circumstances, extensions may be possible if agreed with the Editorial Office in advance. We do not allow multiple rounds of revision so we urge you to make every effort to fully address all of the comments at this stage. If deemed necessary by the Editors, your manuscript will be sent back to one or more of the original reviewers for assessment. If the original reviewers are not available, we may invite new reviewers.

- Data accessibility

<http://datadryad.org/submit?journalID=RSOS&manu=RSOS-190389>

- Competing interests

- Authors' contributions

- Acknowledgements

- Funding statement

on behalf of Professor R. Kerry Rowe (Subject Editor)
openscience@royalsociety.org

Associate Editor's comments:

Please provide a point by point response to each of the reviewers' comments.

Reviewer 2 has recommended that the quality of English should be improved. A number of language polishing services are available for authors whose first language is not English.
<https://royalsociety.org/journals/authors/language-polishing/>

Authors whose papers are returned on language grounds must provide evidence that a professional language editing service or a native speaker of English have assisted in preparing a revised manuscript. Evidence such as a certificate of editing or a signed letter from a native speaker of English would be acceptable.

We look forward to receiving your revised manuscript.

Comments to Author:

Reviewers' Comments to Author:

Reviewer: 1

Comments to the Author(s)

The research was carried out in a comprehensive way which gives sufficient credibility to the main results. The particular value of the paper is in its original approach for analysing the results. Interesting paper with an interesting approach that few researchers considered previously in this field. However, I would like to recommend the following improvements:

1. In the Matlab analysis for the obtained SEM micro-images, authors did not mention the dilemma of noise in the background. How authors managed to reduce the noise in the images for a proper analysis?
2. In the abstract, authors have mentioned the OTSU method. In which part of the research did authors use this method? I could only notice that the authors have used the box-counting method.
3. In section 2.1 (Raw materials), please provide more information about Tailings A and B.
4. The introduction part is very short and needs to be extended. Despite the presence of very few studies that discuss such materials and even the techniques that were used in this research, authors need to explain more about other materials that were used to enhance the performance of cement. Following papers could help:

- A novel approach of introducing crystalline protection material and curing agent in fresh concrete for enhancing hydrophobicity.
- Development of hydrophobic concrete by adding dual-crystalline admixture at mixing stage.
- Influence of early water exposure on modified cementitious coating.

The paper is very interesting and I recommend it for publication but after the authors rectify the comments.

Reviewer: 2

Comments to the Author(s)

The relation between micro porous structure and macro parameters of backfills is difficult to be quantitatively calculated. This paper shares a method through calculating the fractal dimension of pores. It is really an interesting topic.

The major problem is the language. The English needs substantial editing.

So I recommend that this paper can be published after language polish.

Author's Response to Decision Letter for (RSOS-190389.R0)

See Appendix A.

RSOS-190389.R1 (Revision)

Review form: Reviewer 1

Is the manuscript scientifically sound in its present form?

Yes

Are the interpretations and conclusions justified by the results?

Yes

Is the language acceptable?

Yes

Is it clear how to access all supporting data?

Yes

Do you have any ethical concerns with this paper?

No

Have you any concerns about statistical analyses in this paper?

No

Recommendation?

Accept as is

Comments to the Author(s)

Authors have made all the necessary changes. I recommend publishing the paper

Decision letter (RSOS-190389.R1)

10-May-2019

Dear Dr Ren:

On behalf of the Editors, I am pleased to inform you that your Manuscript RSOS-190389.R1 entitled "Trans-scale relationship analysis between the pore structure and macroparameters of backfill and slurry" has been accepted for publication in Royal Society Open Science subject to minor revision in accordance with the referee suggestions. Please find the referees' comments at the end of this email.

The reviewers and Subject Editor have recommended publication, but also suggest some minor revisions to your manuscript. Therefore, I invite you to respond to the comments and revise your manuscript.

- Ethics statement

- Data accessibility

If you wish to submit your supporting data or code to Dryad (<http://datadryad.org/>), or modify your current submission to dryad, please use the following link:
<http://datadryad.org/submit?journalID=RSOS&manu=RSOS-190389.R1>

- Competing interests

- Authors' contributions

- Acknowledgements

- Funding statement

Because the schedule for publication is very tight, it is a condition of publication that you submit the revised version of your manuscript before 19-May-2019. Please note that the revision deadline will expire at 00.00am on this date. If you do not think you will be able to meet this date please let me know immediately.

- 1) A text file of the manuscript (tex, txt, rtf, docx or doc), references, tables (including captions) and figure captions. Do not upload a PDF as your "Main Document".
- 2) A separate electronic file of each figure (EPS or print-quality PDF preferred (either format should be produced directly from original creation package), or original software format)
- 3) Included a 100 word media summary of your paper when requested at submission. Please ensure you have entered correct contact details (email, institution and telephone) in your user account
- 4) Included the raw data to support the claims made in your paper. You can either include your data as electronic supplementary material or upload to a repository and include the relevant doi within your manuscript

5) All supplementary materials accompanying an accepted article will be treated as in their final form. Note that the Royal Society will neither edit nor typeset supplementary material and it will be hosted as provided. Please ensure that the supplementary material includes the paper details where possible (authors, article title, journal name).

on behalf of Prof R. Kerry Rowe (Subject Editor)
openscience@royalsociety.org

Associate Editor Comments to Author:

The reviewer recommends the paper be accepted; however, the Editors asked you to seek language editing (<https://royalsociety.org/journals/authors/language-polishing/>) prior to resubmitting, and there is no evidence that the authors have done this. Please ensure that you seek advice on improving the written language of the paper -- we can only publish once you've done this, and if you submit the revision without evidence from a qualified language editor that the paper has been edited for language, we will return the paper to you.

Reviewer comments to Author:

Reviewer: 1

Comments to the Author(s)

Authors have made all the necessary changes. I recommend publishing the paper

Decision letter (RSOS-190389.R2)

03-Jun-2019

Dear Dr Ren,

I am pleased to inform you that your manuscript entitled "Trans-scale relationship analysis between the pore structure and macroparameters of backfill and slurry" is now accepted for publication in Royal Society Open Science.

on behalf of Prof R. Kerry Rowe (Subject Editor)
openscience@royalsociety.org

Follow Royal Society Publishing on Twitter: [@RSocPublishing](https://twitter.com/RSocPublishing)
Follow Royal Society Publishing on Facebook:
<https://www.facebook.com/RoyalSocietyPublishing.FanPage/>
Read Royal Society Publishing's blog: <https://blogs.royalsociety.org/publishing/>

Appendix A

Details in revision according to reviewer's comments and suggestions on the manuscript "Royal Society Open Science RSOS-190389"

Title of paper: Trans-scale relationship analysis between pore structure and macroparameters of backfill and slurry

Authors: Jianhua Hu¹, Qifan Ren¹, Xiaotian Ding¹, Quan Jiang¹

¹ School of Resources and Safety Engineering, Central South University, Changsha 410083, China

* Correspondence: qifanren@csu.edu.cn

We addressed the comments of the reviewer (with green color) with responses as listed below, corresponding to necessary changes, additional sentences and short sections at proper places which are marked by red color in the revised version of manuscript.

Editor

Q1: Reviewer 2 has recommended that the quality of English should be improved. A number of language polishing services are available for authors whose first language is not English. <https://royalsociety.org/journals/authors/language-polishing/>

Response 4: The manuscript was polished by American Journal Experts. The Editorial Certificate is in the following.

Reviewer: 1

Q1-(Reviewer1): In the Matlab analysis for the obtained SEM micro-images, authors did not mention the dilemma of noise in the background. How authors managed to reduce the noise in the images for a proper analysis?

Response 1: Yes, the noise in background is a dilemma. So I testes many methods to obtain ideal results. The details are as follows. The content has been added in section 3.3 SEM image preprocessing.

3.3 SEM image preprocessing

Image preprocessing mainly includes two aspects of image contrast adjustment and noise reduction.

3.3.1 Image contrast adjustment

Contrast adjustment is the basis of image edge extraction and image segmentation. By adjusting image gray histogram, the color difference between target area and background area can be enhanced. Consequently target features can be enhanced, and background features can be suppressed. This study tested the *decorrstretch* (de-correlation stretching), *adapthisteq* (finite contrast adaptive histogram equalization), *histeq* (histogram equalization), *imadjust*, *stretchilm*,

and *imcontrast* (calling contrast adjustment tool) functions which were used to adjust image contrast. Finally, the *imcontrast* function was used to adjust the image contrast, which can use an image contrast adjustment dialog box to realize the real-time observation adjustment effect and directly extract the optimal adjustment results.

3.3.2 Image noise reduction

Image noise can cause image blurring, pitting, etc., which will directly affect the image binarization results and bring errors to quantitative analysis of images. In this study, the effect of noise on quantitative analysis is reduced by image noise reduction operations and morphological operations after image segmentation. This study tested and compared *medfilt2* two-dimensional median filtering, *ordfilt* two-dimensional sorting statistical filtering, *wiener2* two-dimensional Wiener filtering, *filter2* mean filtering, *DTC* discrete cosine variation, *deconvblind* blind deconvolution operation, a total of six methods about noise reduction. In order to preserve the image details as much as possible during noise reduction, it was finally determined to use the *filter2* mean filtering method (average mean filter) to reduce image noise. The noise reduction operation will cause tailing particles area to be blurred. Therefore, during denoising SEM images of different quality backfill, the parameters should be carefully selected.

Q2-(Reviewer1): In the abstract, authors have mentioned the OTSU method. In which part of the research did authors use this method? I could only notice that the authors have used the box-counting method.

Response 2: the section 3.4 SEM image segmentation uses this method. But there is a mistake of translation. I translated the proper noun “OTSU” to the maximum inter-class variance method in the text. This mistake has been revised in the manuscript.

In section 1 Introduction: The images were processed and the fractal dimensions of SEM images of the microscopic porous structure were calculated using OTSU and box dimension methods.

In section 3.3.2 Grayscale binarization: OTSU is an adaptive method for determining the image segmentation threshold. In MATLAB, OTSU can be called by the function to calculate the image segmentation threshold.

Figure4. The process for calculating the gray threshold of OTSU.

Q3-(Reviewer1): In section 2.1 (Raw materials), please provide more information about Tailings A and B.

Response 3: I have added information about Tailings A and B in the aspects of physical properties and SEM images. The revised section is as follows:

The tailing samples used in the study were obtained from the Gaofeng mine in Guangxi province, China. The collected samples were divided into two types, i.e., Tailings A and Tailings B.

Tailings A is produced by an old-fashioned beneficiation process and Tailings B is produced by a new type of beneficiation in a tin ore. The siliceous limestone was obtained from the quarry around the Gaofeng mine. In this experiment, the samples of mine waste siliceous limestone were ground in a horizontal ball mill with a volume of 35L, whereas the cylinder's rotational speed was 150 r/min. The processing time was 20 minutes. The limestone was made into stone powder having a certain particle size range, which was used to replace part of the cement. The basic physical properties of stone powder and tailings are shown in table 1. P42.5 cement with the strength grade of C30, produced by Changsha Xinxing Cement Factory, was selected as the cementing material. The main elements of the experimental raw materials were analyzed using the Dutch PANalytical X-ray fluorescence r, and the results are presented in table 2. Additionally, Mastersizer2000 was used to analyze the particle size of the raw materials, and the results are shown in figure 1. The SEM images of stone powder and tailings are shown in figure 2.

Table 1. The basic physical properties of stone powder and tailings

Raw materials	Apparent density (g/cm ³)	Packing density (g/cm ³)	Surface moisture content (%)
Tailings A	3.49	1.24	0.128
Tailings B	2.77	1.17	0.135
Stone powder	2.89	0.99	0.162

Figure 2. SEM images of stone powder and tailings; figures (a), (b) and (c) represents Tailings A, Tailings B and stone powder

Q4-(Reviewer1): The introduction part is very short and needs to be extended. Despite the presence of very few studies that discuss such materials and even the techniques that were used in this research, authors need to explain more about other materials that were used to enhance the performance of cement. Following papers could help:

- A novel approach of introducing crystalline protection material and curing agent in fresh concrete for enhancing hydrophobicity.
- Development of hydrophobic concrete by adding dual - crystalline admixture at mixing stage.
- Influence of early water exposure on modified cementitious coating.

Response 4: The introduction part has been extended. In this part other materials that were used to enhance the performance of cement are introduced. The revised introduction and literatures are as follows:

In order to reduce the cost of cement in concrete and to manufacture special cement for various special purposes, many worldwide scholars have conducted extensive work on cement substitute materials and concrete additives. Siddique [1] studied effects of volcanic ash on consistency, setting times, workability, compressive strength, electrical resistivity of cement paste and mortar. Weerdt et al. [2] found that the presence of 5% of limestone in concrete led to an increase of the volume of the hydrates, as visible in the increase in chemical shrinkage, and an increase in compressive strength. Kupwade-Patil et al. [3] investigated the effectiveness of the use of volcanic ash along with silica fume as a partial replacement for Portland cement. Celik et al. [4] reported the composition and properties of highly flowable self-consolidating concrete mixtures made of high proportions of cement replacement materials such as fly ash and pulverized limestone instead of high dosage. Al-Kheetan et al. [5-7] introduced crystalline material along with a curing compound in fresh concrete to protect and extend its service life and developed hydrophobic concrete by adding dual-crystalline admixture during the mixing stage. However, there are a few materials used in mines backfilling and the research on these materials is limited. Hu et al. [8] assessed stone powder as a replacement of cement in cemented paste backfill (CPB), and studied strength characteristics and reaction mechanism. The results showed that strength of the backfill greatly reduced at an early stage, and slightly reduced in the final stages, while trans-scale characteristics between pore structure and macroparameters are still needed to be studied.

- [1] Siddique R. 2012 Properties of concrete made with volcanic ash. *Resour. Conserv. Recycl.* 66, 40-44. (doi.org/10.1016/j.resconrec.2012.06.010)
- [2] De Weerdt K, Ben Haha M, Le Saout G, Kjellsen KO, Justnes H, Lothenbach B. 2011 Hydration mechanisms of ternary Portland cements containing limestone powder and fly ash. *Cem. Concr. Res.* 41, 279-291. (doi.org/10.1016/j.cemconres.2010.11.014)
- [3] Kupwade-Patil K, Palkovic SD, Bumajdad A, Soriano C, Buyukozturk O. 2018 Use of silica fume and natural volcanic ash as a replacement to Portland cement: Micro and pore structural investigation using NMR, XRD, FTIR and X-ray microtomography. *Constr. Build. Mater.* 58, 574-590. (doi.org/10.1016/j.conbuildmat.2017.09.165)
- [4] Celik K, Meral C, Gursel AP, Mehta PK, Horvath A, Monteiro PJM. 2015 Mechanical properties, durability, and life-cycle assessment of self-consolidating concrete mixtures made with blended portland cements containing fly ash and limestone powder. *Cem. Concr. Compos.* 56, 59-72. (doi.org/10.1016/j.cemconcomp.2014.11.003)
- [5] Al-Kheetan MJ, Rahman MM, Chamberlain DA. 2018 A novel approach of introducing crystalline protection material and curing agent in fresh concrete for enhancing hydrophobicity. *Constr. Build. Mater.* 160, 644-652. (doi.org/10.1016/j.conbuildmat.2017.11.108)
- [6] Al-Kheetan MJ, Rahman MM, Chamberlain DA. 2018 Development of hydrophobic concrete by adding dualcrystalline admixture at mixing stage. *Struct. Concr.* 19, 1504-1511. (doi.org/10.1002/suco.201700254)
- [7] Al-Kheetan MJ, Rahman MM, Chamberlain DA. 2017 Chamberlain. Influence of early water exposure on modified cementitious coating. *Constr. Build. Mater.* 141, 64-71.

- [8] Hu JH, Ren QF, Jiang Q, Gao RG, Zhang L, Luo ZQ. 2018 Strength characteristics and the reaction mechanism of stone powder cement tailings backfill. *Adv. Mater. Sci. Eng.* 2018, 1-14. (doi.org/10.1155/2018/8651239)
- [9] Shang JL, Hu JH, Zhou KP, Luo XW, and Aliyu M. 2015 Porosity increment and strength degradation of low-porosity sedimentary rocks under different loading conditions. *Int. J. Rock Mech. Min. Sci.* 75, 216-223. (dx.doi.org/10.1016/j.ijrmms.2015.02.002)
- [10] Zhou KP, Gao RG, Gao F. 2017 Particle Flow Characteristics and Transportation Optimization of Superfine Unclassified Backfilling. *Minerals.* 7, 6. (doi.org/10.3390/min7010006)

Reviewer: 2

Q1-(Reviewer2): The relation between micro porous structure and macro parameters of backfills is difficult to be quantitatively calculated. This paper shares a method through calculating the fractal dimension of pores. It is really an interesting topic. The major problem is the language. The English needs substantial editing.

So I recommend that this paper can be published after language polish.

Response 1: Yes I seriously and carefully revised the language of this manuscript. Following this, the manuscript was polished by a professional language polishing agency and a native English editor. The Certificate of English Editing is in the following.

EDITORIAL CERTIFICATE

This document certifies that the manuscript listed below was edited for proper English language, grammar, punctuation, spelling, and overall style by one or more of the highly qualified native English speaking editors at American Journal Experts.

Manuscript title:

Trans-scale relationship analysis between pore structure and macroparameters of backfill and slurry

Authors:

Jianhua Hu, Qifan Ren , Xiaotian Ding, Quan Jiang

Date Issued:

April 23, 2019

Certificate Verification Key:

4B8D-48DD-FEC0-C7DC-5DF5

This certificate may be verified at www.aje.com/certificate. This document certifies that the manuscript listed above was edited for proper English language, grammar, punctuation, spelling, and overall style by one or more of the highly qualified native English speaking editors at American Journal Experts. Neither the research content nor the authors' intentions were altered in any way during the editing process. Documents receiving this certification should be English-ready for publication; however, the author has the ability to accept or reject our suggestions and changes. To verify the final AJE edited version, please visit our verification page. If you have any questions or concerns about this edited document, please contact American Journal Experts at support@aje.com.